# Improving Access to Radiotherapy: Exploring Structural Quality Indicators for Radiotherapy in Gauteng Province, South Africa

**DOI:** 10.3390/ijerph22040585

**Published:** 2025-04-08

**Authors:** Portia N. Ramashia, Pauline B. Nkosi, Thokozani P. Mbonane

**Affiliations:** 1Department of Environmental Health, Faculty of Health Sciences, University of Johannesburg, Johannesburg 2000, South Africa; tmbonane@uj.ac.za; 2Faculty of Health Sciences, Durban University of Technology, Durban 4000, South Africa; paulinen1@dut.ac.za

**Keywords:** radiotherapy, access, quality indicators, Gauteng province, South Africa, cancer care, healthcare system, mixed methods, resource allocation, health equity

## Abstract

Background: Radiotherapy is a critical component of effective cancer treatment, yet access remains limited in many low- and middle-income countries, including South Africa. This study explores structural quality indicators influencing radiotherapy access in Gauteng province, a region with a dual public–private healthcare system. Methods: A concurrent triangulation mixed-methods approach using a descriptive cross-sectional study was employed; for the quantitative phase, data from public and private radiotherapy facilities was analysed, and for the qualitative phase, insights were obtained from interviews with the heads of departments and members of three multidisciplinary professions in radiation oncology, namely radiation oncologists, radiation therapists, and medical physicists. Results: Findings reveal stark disparities in radiotherapy access. Gauteng province has only two major public radiotherapy facilities serving a large population, while multiple private facilities exist. The data indicate substantial differences in resource distribution, equipment accessibility, and personnel levels between public and private institutions. Bureaucratic inefficiencies, personnel shortages, and operational limitations in the public sector have surfaced as significant barriers to prompt equitable access to radiotherapy. This research shows the urgent need for focused strategies to address these systemic issues in order to improve access to radiation treatment in Gauteng province. The study’s findings will inform the development of a comprehensive framework to enhance radiotherapy access and contribute to more equitable cancer care delivery in South Africa.

## 1. Introduction

Radiotherapy is a critical component of effective cancer treatment, often playing a vital role in curative and palliative care strategies. Despite its importance, access to radiotherapy remains severely limited in many parts of the world, particularly in low- and middle-income countries [1]. The African continent, for instance, faces a significant disparity in radiotherapy access, with a considerably lower number of radiotherapy machines per capita compared to high-income regions [2,3]. This disparity contributes to poorer cancer outcomes and underscores the urgent need to improve access to radiotherapy services across Africa [4].

The availability of adequately trained and qualified personnel, including Radiation Oncologists (ROs), Radiation Therapists (RTTs), Medical Physicists (MPs), and Oncology Nurses (ONs), is crucial for workforce effectiveness [5,6]. Indicators may assess the number of staff per patient volume or per equipment unit [6]. Access to modern and well-maintained radiotherapy equipment, such as linear accelerators, brachytherapy units, treatment planning systems, and quality assurance tools, is essential, with indicators considering the age and type of equipment, maintenance records, and availability of advanced technology [4,7]. Appropriate infrastructure, including treatment rooms, simulation facilities, waiting areas, and support services, contributes to a safe and efficient environment, with indicators evaluating facility size, layout, safety features, and patient comfort. Robust information technology systems are vital for managing patient data, treatment planning, dose delivery, and quality assurance, with indicators assessing the use of electronic health records, treatment planning software, and data analysis tools. The presence of comprehensive quality management programmes ensures consistent adherence to protocols, safety standards, and continuous improvement efforts, with indicators evaluating the implementation of quality assurance procedures, incident reporting mechanisms, and staff training programmes [5].

Structural quality indicators in radiotherapy assess foundational elements that influence the delivery of high-quality care, focusing on tangible resources, organisational aspects, and infrastructure rather than processes or outcomes [6]. Key indicators include workforce availability, which evaluates the number of adequately trained personnel per patient volume or equipment unit; equipment access, which considers the age, type, and maintenance of radiotherapy tools; facilities, which assess infrastructure aspects like treatment rooms and patient comfort; information systems, which evaluate the use of technology for managing patient data and treatment planning; and quality management systems, which ensure adherence to protocols and continuous improvement through quality assurance procedures and staff training. By analysing these structural quality indicators, policymakers and healthcare providers can identify areas for improvement, allocate resources effectively, and ultimately enhance the quality and accessibility of radiotherapy services [8].

This study represents one phase of a larger research project designed to develop a comprehensive, evidence-based strategic framework for improving radiotherapy access for cancer patients in Gauteng province, South Africa. This phase examines the landscape of radiotherapy access in Gauteng province, South Africa, a region characterised by a dual public and private healthcare system, while the other phases looked at quantitative data on time intervals and patient experiences. We aim to explore the structural quality indicators of radiotherapy services and their impact on the access for cancer patients. By analysing data from both public and private radiotherapy facilities, this study provides a comprehensive assessment of the radiotherapy landscape in Gauteng province. Furthermore, we investigate the potential role of the private sector in supplementing state-provided services to improve overall access to radiotherapy for cancer patients. This research employs a concurrent triangulation mixed-methods approach, combining quantitative data analysis from radiotherapy facilities with qualitative insights from interviews conducted with key stakeholders, including radiation oncologists, radiation therapists, and medical physicists. This multifaceted approach allows for a nuanced understanding of the factors influencing radiotherapy access and quality in Gauteng province. The findings of this phase of the study will be instrumental in developing a comprehensive framework to improve access to radiotherapy by cancer patients in Gauteng province, South Africa, which is the goal of the postgraduate study that this phase is part of.

## 2. Materials and Methods

### 2.1. Study Approach and Design

This study employed a concurrent triangulation mixed-methods approach, while a descriptive cross-sectional design was used to explore the structural quality indicators of radiotherapy services in Gauteng province, South Africa, and their impact on patient access to radiotherapy. Data were collected from facilities’ statistical data for the quantitative phase and interviews for the qualitative phase. The study was conducted in a month-to-month manner.

### 2.2. Study Setting and Population

The study employed a mixed-methods approach, combining quantitative data analysis from both public and private radiotherapy facilities with qualitative insights from interviews. While quantitative data were collected from a mix of public and private facilities, the qualitative phase focused specifically on gathering perspectives from healthcare professionals at two public radiotherapy facilities in Gauteng province. This focus allowed for a deeper exploration of the challenges and opportunities within the public sector context. For the quantitative phase, the facilities’ data were acquired from heads of the radiation oncology departments (HODs) of radiotherapy departments. For the qualitative phase, the participants were HODs of the radiation oncology departments; the radiation oncologists (ROs), medical physicists (MPs), and radiation therapists (RTTs) in the public facilities.

### 2.3. Sampling

While the intention was to include all radiotherapy practices in Gauteng province (2 public and 20 private), logistical constraints resulted in incomplete participation. We successfully recruited 13 practices (2 public and 11 private). Six private practices were excluded due to delays in obtaining gatekeepers’ permission, one practice closed during the data collection period, and another declined participation due to a busy schedule. This deviates from total population sampling, introducing the potential for bias. The practices that did not grant permission might have differed systematically from those that did. Similarly, the closure of one practice could indicate other issues not representative of the included practices, and the practice declining due to a busy schedule might have had a higher patient volume. Due to these limitations, the findings of this study should be interpreted with caution, and future research should employ more comprehensive sampling strategies to ensure greater representativeness.

### 2.4. Data Collection Approach

The study collected both quantitative and qualitative data to achieve the study’s aim and objectives. Quantitative data were collected using a record review tool, while face-to-face interviews were used for qualitative data.

#### 2.4.1. Quantitative Phase (Record Review)

The collected quantitative data included information on staffing, equipment, and the number of patients treated and waiting to start radiotherapy. The data collection tool was adapted from the International Atomic Energy Agency (IAEA) quality indicators tool (Table 1).

#### 2.4.2. Qualitative Phase (Interview)

Qualitative data were collected from two public radiotherapy facilities in Gauteng province through face-to-face interviews with the HODs of the radiation oncology departments and ROs, MPs, and RTTs. The intention was to also include the heads of ONs, but they were both not available. The interviews explored the challenges faced in providing radiotherapy services, including barriers related to staffing, training, and equipment. This approach provided a comprehensive understanding of the challenges and opportunities related to radiotherapy access from the perspectives of key personnel (ROs, MPs, and RTTs) involved in radiotherapy delivery.

### 2.5. High-Energy Units and Workload Estimation

The availability and utilisation of high-energy units (HEUs), such as linear accelerators, are critical factors in determining radiotherapy access. To assess HEU utilisation, the utilisation ratio was calculated using the following formula:(1)$$\mathrm{Use}\;\mathrm{of}\;\mathrm{high}\text{-}\mathrm{energy}\;\mathrm{units}=\frac{Total\;number\;of\;patients\;treated\;in\;1\;year}{Number\;of\;high\;energy\;units}$$

This ratio reflects the average number of patients treated per HEU annually, providing insights into equipment efficiency and potential resource strain. A higher ratio may indicate higher efficiency or, conversely, potential overutilisation and strain on resources.

To evaluate the workload of radiation therapy professionals, the following formula was used, adapted from international standards:(2)$$\mathrm{Workload}=\frac{Total\;number\;of\;patients\;treated\;in\;1\;year}{Number\;of\;workers}$$

This calculation compares workload distribution across different professional categories (radiation oncologists, therapists, and physicists) and between the public and private sectors. By comparing calculated workloads to established benchmarks or guidelines, the staffing levels were assessed to determine whether they were adequate to meet patient demand and maintain quality care. This analysis also informed the broader investigation into resource allocation and its impact on radiotherapy access in Gauteng province.

### 2.6. Data Analysis

#### 2.6.1. Quantitative Phase Analysis

Data were entered into Microsoft Excel and then analysed using IBM SPSS Version 29 software. Frequencies and descriptive statistics were used to analyse and calculate key quality indicators of radiotherapy, including the staff-to-equipment-to-patient ratio and the average number of patients waiting to start treatment. Categorical variables were summarised with counts and percentages for the ratios; median, minimum, and maximum values were used. The statistics were run for the entire population and then were split by facility type (public or private).

#### 2.6.2. Qualitative Phase Analysis

The qualitative interviews were transcribed and analysed using a thematic analysis approach to explore the key challenges and barriers to radiotherapy access as perceived by the HODs and key stakeholder professionals in radiotherapy departments.

### 2.7. Ethical Considerations

The relevant institutional ethics review board approved the study, and all participants provided informed consent before they participated in the study.

## 3. Results

This study provides a comprehensive assessment of structural quality indicators in radiotherapy services in Gauteng province. This section integrates these findings to provide a nuanced perspective on the factors influencing access to and quality of radiotherapy services. Employing a concurrent triangulation mixed-methods approach, the results reveal notable disparities in radiotherapy access between public and private facilities. Key themes that emerged from this integrated analysis include limited radiotherapy capacity impacting waiting times and treatment efficacy, disparities between the public and private sectors affecting technology availability and resource allocation, and systemic issues such as bureaucratic inefficiencies and communication breakdowns. These findings highlight the need for comprehensive strategies that address both structural and systemic barriers to ensure equitable access to quality radiotherapy for all patients in Gauteng. The following sections will present the quantitative data that highlight these disparities, followed by qualitative insights from interviews with the heads of the radiation oncology departments in the public facilities, which provide context and explain the underlying factors contributing to these challenges.

### 3.1. Quantitative Phase Results

This quantitative analysis aims to provide a comprehensive view of the radiotherapy landscape in Gauteng province, South Africa, by examining key characteristics of both public and private radiotherapy facilities. The results reveal notable differences in patient volume, equipment availability, and staffing levels between the two sectors. As shown in Table 2, the two public radiotherapy facilities in this study had a substantially higher average number of patients per day (mean = 102.50) compared to the private facilities (mean = 31.18) [1,4,9,10]. This noticeable disparity in patient load between the public and private facilities suggests a potential strain on resources and service provision within the public radiotherapy sector.

Significant variations were also observed in the availability of critical radiotherapy equipment. While all facilities reported having at least one CT simulator, the analysis of the data revealed that brachytherapy units, essential for internal radiation therapy, were present in both public facilities but only available in a limited capacity within the private sector. Similarly, access to MRI machines, which facilitate advanced treatment planning and tumour visualisation, was more prevalent in public facilities than private facilities [1]. This limited capacity is further exacerbated by systemic challenges such as bureaucratic delays in procuring new equipment, as reported in interviews, and insufficient investment in infrastructure, leading to longer waiting times and potentially compromising treatment efficacy.

This difference highlights the importance of considering not just the total number of machines but also their availability relative to the population they serve. To further illustrate this point, our analysis revealed that public radiotherapy facilities had a higher average number of linear accelerators compared to private centres. Specifically, there were 4.5 linear accelerators in the public sector and 1.27 in the private sector in average per facility. However, to account for differences in patient populations served, we calculated the number of linear accelerators per 1000 patients. This normalised metric showed that the public sector had 2.09 linear accelerators per 1000 patients, while the private sector had 2.31 linear accelerators per 1000 patients. This difference in linear accelerator availability, as highlighted by the normalised metric, is reflected in the HEU utilisation ratios presented in Table 3 and the normalised linear accelerator availability in public and private sectors provided in Table 4. These disparities are perpetuated by policy gaps and governance issues that result in unequal funding allocations in public hospitals, as indicated by qualitative data.(3)$$Linear\;Accelerator\;per\;patients=\frac{number\;of\;linear\;accelerators}{total\;patients}\times 1000$$

The workload of radiation oncologists, therapists, and medical physicists in both public and private facilities was analysed to examine the impact of patient volume on resource allocation and utilisation. As shown in Table 5, the workload is calculated as the average number of patients treated per year for each radiation oncologist, therapist, and medical physicist. Public facilities exhibit a considerably higher average workload for radiation oncologists (540–1048 patients/year) compared to private facilities (60–201 patients/year). This disparity is particularly striking when considering the recommended workload ranges of 250–300 patients per year for radiation oncologists. The radiation therapists in workload public and private facilities ranged between 93 and 103 patients/year and 55–115 patients/year, respectively; these ranges are within the recommended range of 100–150 patients per year. However, although the ranges are within the recommended range, it is important to note that these workload figures may be higher than currently represented due to variations in the scope of responsibilities for radiation therapists across the different centre types. For instance, therapists in public facilities are often required to work on a broader range of equipment, including brachytherapy units, which are less common in private facilities. These findings underline the significant strain placed on healthcare professionals within the public sector due to the higher patient volume.

### 3.2. Qualitative Phase Findings

The interviews with the heads of the radiotherapy departments from the two public facilities echo the quantitative results. The thematic analysis of the qualitative data highlights specific barriers related to limited radiotherapy facilities, government bureaucracy and tendering challenges, staff shortages and remuneration disparities, and operational challenges. These systemic issues further exacerbate the challenges faced by public radiotherapy facilities and hinder their ability to provide timely and effective care, resulting in longer waiting times for treatment in the public sector, potentially leading to disease progression and poorer outcomes and widening the health equity gap between different socioeconomic groups in the province.

#### 3.2.1. Limited Radiotherapy Facilities

Limited facilities have been expressed as a considerable challenge and limitation to radiotherapy. The radiation oncology healthcare professionals mentioned that there are limited treatment facilities for cancer patients, with only two facilities in Gauteng province accommodating many patients. The limited facilities then result in delays in receiving radiation therapy within the recommended timeframe. As the quantitative data indicate, the higher patient load per machine in the public sector likely contributes to these delays. The delay is further compounded by the fact that the existing facilities also serve patients from other provinces without radiotherapy centres. As is seen in the following excerpt:

“*The major issue here in Gauteng province is the population that we have. We have many patients diagnosed with cancer, and we only have two facilities, for therefore our patients are not going to be getting radiation within the recommended period*”(CMJAH Healthcare Professional 1).

“*Okay, so Gauteng province has only two major academic facilities which provide healthcare to the majority of the population in South Africa. So, 85% of the patients do not have access to medical services and rely on the public health system*”(SBAH Healthcare Professional 3).

This lack of resources is not merely a matter of funding, it is also tied to systemic issues such as bureaucratic inefficiencies and a lack of coordinated planning, further straining the capacity of the department to provide timely and effective care.

#### 3.2.2. Government Bureaucracy and Tendering Challenges

Inefficient bureaucratic processes and tendering issues create significant obstacles for public radiotherapy facilities. Tender and procurement processes were expressed as another challenge, which often impacted the timeline for equipment used in treating cancer patients. The payment process in the procurement cycle can be a significant challenge due to delays and bureaucratic red tape, causing frustration and inefficiencies. There are challenges with equipment repairs and payment of invoices. The payment process is seen as overly complicated due to multiple levels of interference and tender rules. The process involves requesting quotes, creating purchase orders, and receiving services or goods from suppliers. However, delays often occur when submitting documents for purchase order numbers or navigating complex procurement procedures. These challenges contribute to extended timelines for acquiring equipment or services, impacting operational efficiency. For instance, malfunctioning machines cause delays at a radiotherapy facility. These challenges cause prolonged waiting times due to machines being down and limited in number. As seen in the following quotes:

“*…there’s an issue in planning, we need a solution around how planning needs to work, we need to sort of solve the issue regarding staffing and getting that, you know … So it wasn’t difficult things. Unfortunately, it’s a bureaucracy that is very slow moving to try and get any of those things done. So it was very simple to diagnose what the issue was, but then to actually get people actioning any of this is a very long process*”(CMJAH Healthcare Professional 2).

“*So, even getting the two compact Linacs onto the tender, which I started when I started here in 2021, only ended at the beginning of this year. The Brachytherapy was even longer, that was a five-year tendering process. That started before I even arrived here and only finished at the beginning of this year*”(CMJAH Healthcare Professional 2).

“*Even though the tendering system, the idea was to empower those ones who want to initiate their businesses. The bureaucracy that comes with all these things do need to be looked at. But… The tenders are awarded to people who are not…I do not know whether I should say competent. Or who do not have products they can supply. They become third party.*”(SBAH Healthcare Professional 1).

“*So, the delay arises when you submit the required documents to say we are requesting that we get a PO for the service provider to be able to come and attend to the problem arising. So that, to me, has been an issue …*”(SBAH Healthcare Professional 1).

‪“*So, I think, yeah, I think in terms of equipment as well. I think procurement processes are extremely laborious and painful as well. When I started, I was involved as a chairperson of the steering committee for the acquisition of Brachytherapy for the province. We started the process in October 2019. It was only completed this year, at the beginning of the year, in 2024, and that is because the whole process is just so complex*”(SBAH Healthcare Professional 3).

#### 3.2.3. Staff Shortages and Remuneration Disparities

Staff shortages, driven by remuneration disparities and challenging working conditions, emerged as a critical barrier to providing timely and effective radiotherapy care. These shortages are supported by quantitative data indicating that radiation oncologists in public facilities handle a considerably higher average workload (540–1048 patients/year) compared to their counterparts in private facilities (60–201 patients/year), exacerbating the strain caused by inadequate compensation and challenging working conditions. Healthcare professionals expressed that specialists were underpaid compared to other provinces, mainly due to an occupational-specific dispensation (OSD) introduced by the Department of Health, specifically in the Gauteng province. Therefore, participants expressed a need for healthcare professionals working in specialised areas to be paid accordingly. As seen in the following quote:

“*… the remuneration of the staff in Gauteng province, not CMJAH alone, is different from other provinces. Therapists here are still paid as diagnostic radiographers. I do not know why things are done that way because there is this; I do not know whether it is a policy or what that says equal pay for equal jobs, which Gauteng province health is not practising. So, if these therapists are working in a speciality, they should be paid as specialists*”(CMJAH Healthcare Professional 1).

The remuneration issues then result in specialists seeking employment opportunities in the private sector or other provinces, as stated by two other professionals:

“*… because this is what has happened here, where the biggest challenge was with the therapist, and they were leaving because of this payment issue, right, but in truth, staffing issues are just a problem throughout the specialities*”(SBAH Healthcare Professional 3).

“*So, all these potential workers that wanted to stay with us go back to the private sector, and I think we are going to run into problems because eventually, this private sector is going to be totally overly, they are going to be full*”(SBAH Healthcare Professional 2).

The staff shortages do not only apply to radiation therapists. There is not enough incentive for all qualified specialists in the government sector. The challenging work environment and lack of financial rewards are discouraging for staff who have dedicated years to their studies and careers. The current criteria for radiation therapists do not align with the qualifications of recent graduates, leaving them without entry-level job opportunities. Some provinces have adjusted their rules, but Gauteng Province Health has not. Despite promises to address the issue, it has not been resolved, leading to a situation where qualified individuals cannot secure positions due to outdated regulations. This payment issue has led to staff leaving, affecting all specialities in the facility. Challenges with remuneration and working conditions have led to an imbalance in relation to available healthcare professionals who are available to treat patients. The remuneration and working conditions are part of the reason for the long waiting list. Healthcare professionals leaving the public sectors in Gauteng province and seeking opportunities in other provinces result in challenges with limited staff members, which then impacts the delivery of service. Some participants have expressed that, ultimately, there is a need for more specialists in South Africa since there are only a few available. The following quotes from healthcare professionals support this:

“*… It can be easily resolved. We have tried numerous times with the Department of Health. There has been a memo sent out by Western Cape and KZN … You know, they have amended their rules, even though DPSA has not amended theirs, and said that they can recruit people who do not have the OSD match requirements, even though they have a four-year degree. However, Gauteng Health refuses to do that. My personal opinion, I think it is maybe because of trade unions and because it might spill over into other disciplines where similar problems exist*”(SBAH Healthcare Professional 3).

“*The second thing, I think, is that we do have a staffing issue. So if you are to compare us to the IAEA requirements for staffing, and that is also outdated, you know, and has not been updated, I think we are way, way under par. … there is one full-time consultant, and then there are two doctors who come in to do one-day sessions. One of them might be leaving, so that is one thing. Then, two registrars are qualifying and will soon have to register as specialists. Nevertheless, the biggest thing is that they do not have any incentive to stay because the public sector does not pay lucrative salaries*”.(SBAH Healthcare Professional 2).

“*And so if you had to compare what they could earn in the private sector basically, they could earn one month’s salary by just treating five patients, excluding overheads and stuff like that. So I think it is not inviting for them because, you know, the environment is frustrating because there are so many challenges. And then they basically, it is not financially lucrative, and they have studied a long time to get there*”(SBAH Healthcare Professional 2).

“*As we speak, we do not have therapists that can operate all the linear accelerators. So, in my view, if we had enough staff to operate the equipment, we were going to be able to make a difference in these patients that are waiting for treatment*”(CMJAH Healthcare Professional 1).

‪“*… but the biggest challenge currently is the lack of staff to be able to treat all those radiation patients. So, we have five fully functioning linear accelerators currently, but not enough staff to operate all the machines. Up until the beginning of July, we only worked three of our linear, linear accelerators, and then we have like three fully qualified with a comm serve, helping out sometimes after the school holidays*”(SBAH Healthcare Professional 2).

Addressing the human resource issue is crucial. Staff shortages are causing delays in patient treatment and limiting operational effectiveness. Staffing inadequacies also hinder the ability to conduct research and develop local policies. It was emphasised that having skilled individuals is as essential as having equipment.

#### 3.2.4. Operational Challenges

Healthcare professionals highlighted challenges in the workflow and communication within their radiation oncology department. They mentioned the need for improved multidisciplinary teamwork, more structured protocols, better communication between staff and management, and involving all stakeholders in decision-making. They also emphasised the importance of collaborative planning involving therapists, doctors, and medical physicists to streamline processes and ensure efficient patient care. The following quotes support this:

“*The other thing is communicating needs from your end user, I mean us now, and the executive or, you know, that communication sometimes is lost, like, for instance, I will make an example. Say the head of oncology, who is the head of the department, goes and sits with whoever is there, and they say, this is what is needed. However, the problem really is that in oncology, unfortunately, it is a multidisciplinary team, so the communication among professionals from all disciplines should be something that is really addressed*”(CMJAH Healthcare Professional 3).

‪“*And the other one, as I am thinking, would be teamwork if the planners, not planners alone, planners, doctors and medical physics. If we can work together as a team and do these patients together, it will speed up the process. Furthermore, you know, when we work together as a team, even if communication improves, then we will enjoy our work as currently, the way I see things, there are lines drawn between the therapist, the doctor and the medical physicist. More especially between therapists and medical physicists, you will find that when a plan has some challenges, or maybe the medical physics is putting something on a plane, the medical physicist will bring the file back to the therapist instead of communicating directly with the consultant involved in order to resolve that issue, which then delays the treatment of a patient. So, I strongly feel that we need to improve in that as well*”(CMJAH Healthcare Professional 1).

“*It feels like we are all working in silos, and that is not good for patients*”(CMJAH Healthcare Professional 3).

“*I think there will always be a little strain between the radiographers and physicists because, I cannot say it, but it always seems like physics think they are more clever than radiographers, but it is not really the case. Also, because of the strain, we are constantly putting out fires because of poor communication between the teams … As a team, it is a collaboration we all bring to the table to get this patient’s plan planned*”(SBAH Healthcare Professional 2).

Collaborative work was considered necessary even between radiography facilities in the province, as participants deemed it necessary for effective patient treatment. One healthcare professional believed that collaborative work could mean learning from each other. As seen in the following quote:

“*Even though, when we went to Steve Biko during the fire, there were things that our doctors would say no to. Furthermore, Steve Biko’s doctors were doing things like Steve Biko; they would give one single shot of 8Gy, and the patient goes, and here our doctors felt no, no, no with 8Gy single shot is too much for the patient. ‪We want to give it bits by bits. So, such things. If then, that will make both centres consultants get together and they come up with one plan*”(CMJAH Healthcare Professional 1).‬

Efficiency in the referral process can help reduce delays and streamline patient care, as discussed in the following quote:

‪“*They face the same the same issues. But what I would say is things that they can control, like when they refer patients, try to refer the patients, quickly to us, and then the other thing is, when they refer patients to us, to send everything that the patient needs, needs to come with biopsy, scans. Because what happens sometimes is that when the patients come without that, the patients will stay the entire day here, see, wait to see the doctor. The doctor will then write a letter back to their referring doctor and say, “Please, can you send the whatever, you know, CT scan of this patient, or whatever” right? Or, please, can you do a CT scan of this patient? It’s, you know, and it’s frustrating for the patients as well, going up. So, I think if those things that the doctors could just be a bit better*”(CMJAH Healthcare Professional 2).

Healthcare professionals discussed the challenges in planning and treating participants in the medical system, proposing a centralised hub with satellite facilities for oncology and diagnostics. They suggested creating a system to streamline communication and information sharing between facilities to avoid the duplication of services and improve efficiency. Participants also discussed the importance of proper information transfer between healthcare providers for effective patient care.

“*I think, because it sounds like a Steve Biko, they do not have a long waiting period like us. Maybe if we can have a pool where we’re all going to take from it to treat these patients, it might help. So a pool, we just have, we have a centralised planning system, so to say, this is the patient that has just been diagnosed, … then we all take from that pool to plan the patient, and it goes to the machine. Because I understand Steve Biko has some patients that are waiting to be put on the machine, and with us, we do not have such patients. Instead, we are having a challenge in terms of planning*”(CMJAH Healthcare Professional 1).

“*… However, what I would say is things that they can control, like when they refer patients, try to refer the patients quickly to us, and then the other thing is, when they refer patients to us, to send everything that the patient needs, needs to come with biopsy, scans. What happens sometimes is that when the patients come without that, the patients will stay the entire day here, waiting to see the doctor. The doctor will then write a letter back to their referring doctor and say, “Please, can you send the whatever … you know…, a CT scan of this patient, it is frustrating for the patients as well, going up and down. So, I think if those things are addressed, the doctors could be a bit better*”(CMJAH Healthcare Professional 2).

## 4. Discussion

This study examined radiotherapy access in Gauteng province, South Africa, revealing significant disparities between the public and private sectors. The quantitative analysis of high-energy unit utilisation demonstrated higher workloads and a potential strain on resources in public facilities compared to private facilities. Furthermore, workload distribution analysis revealed substantial variations among radiotherapy professionals, with some exceeding recommended standards while others fell short. Qualitative findings corroborated these results, highlighting key barriers to timely and effective radiotherapy services, including limited facilities, bureaucratic tendering processes for equipment acquisition, staff shortages with remuneration disparities, and operational challenges such as equipment breakdowns and inadequate maintenance. These systemic issues exacerbate the challenges faced by public radiotherapy facilities, contributing to longer waiting times [4,11]. Addressing these systemic issues will require a comprehensive, multifaceted approach involving policy reforms, targeted investments, and collaborative efforts between the public and private sectors.

Another key challenge is the stark imbalance in the distribution of facilities between the public and private sectors. There are only two public radiotherapy facilities compared to 20 private facilities serving a population of 15.83 million [12,13]. These results translate to a population-to-facility ratio of 7.9 million people per centre in the public sector. This disparity is further compounded by variations in radiotherapy capacity within each sector. Public facilities, on average, operate with four linear accelerators and one brachytherapy unit staffed with 22 radiotherapists.

In contrast, private facilities, on average, have 1.27 linear accelerators and 0.27 brachytherapy units, with 2.27 radiotherapists. These figures underscore the significant high patient load and potential resource strain in public facilities. Benchmarking these ratios against international standards, such as the IAEA recommendation of one megavoltage machine per 250,000–400,000 population, reveals a substantial shortfall in radiotherapy capacity within Gauteng province, particularly in the public sector [1]. This scarcity of resources contributes directly to extended waiting times and potentially compromises the quality of care delivered.

The limited radiotherapy facilities observed in Gauteng province echo a global trend, particularly pronounced in LMICs, where access to radiotherapy remains a significant challenge [14,15,16]. The findings of this study indicate a potential shortage of radiotherapy resources in Gauteng province. For context, it is important to consider the number of linear accelerators available relative to the patient population. In Gauteng province, approximately 4308 patients receive radiotherapy treatment annually. With a total of 9 linear accelerators (of which 6 are currently in use), this translates to 2.09 linear accelerators per 1000 patients when considering all machines and 1.39 linear accelerators per 1000 patients when considering only those in operation. This falls slightly short of the recommendation of 2.2 linear accelerators per 1000 patients, highlighting a need to maintain the current infrastructure to meet the demands of the population. The number of machines needed can be calculated based on how many fractions can be delivered per machine per year divided by the number of fractions per patient [1]. While public and private facilities may serve different patient populations, the resource imbalance clearly indicates a need for strategic investment and resource allocation to ensure equitable access to timely and effective radiotherapy services across the province.

A critical constraint on radiotherapy service delivery in Gauteng province is the severe shortage of qualified personnel, particularly within the public sector. This scarcity spans across key roles, including radiation oncologists, medical physicists, and radiation therapists. Interview data revealed a significant disparity in remuneration between the public and private sectors for these positions. Healthcare professionals in the public sector frequently cited lower salaries and less favourable working conditions compared to their private sector counterparts, leading to an exodus of skilled personnel to private institutions or other provinces. This trend has resulted in increased workload and burnout among remaining staff in the public sector, ultimately impacting the quality of patient care. The brain drain further exacerbates the strain on the already limited public radiotherapy services, resulting in increased workloads for remaining staff, potential compromises in patient care, and extended waiting times for treatment. In addressing the critical workforce shortage, it is essential to implement strategies that will attract and retain qualified professionals in public facilities; competitive salaries and benefits packages are necessary. Furthermore, to mitigate the impact of these workforce shortages and improve the efficiency of radiotherapy service delivery, exploring the use of artificial intelligence (AI) technologies might be beneficial. AI can automate tasks such as contouring or treatment planning and optimise treatment plans and workflows [17,18].

While the shortage of qualified personnel poses a significant challenge, these human resource issues are often exacerbated by bureaucratic hurdles and inefficiencies in the procurement and maintenance of essential radiotherapy equipment. As Ambe notes in a study on the role of public procurement to socioeconomic development, “*public procurement activities suffer from neglect, a lack of direction, poor co-ordination, a lack of open competition and transparency, differing levels of corruption, and most importantly, not having a cadre of trained and qualified procurement specialists who can conduct and manage procurement in a professional, timely and cost-effective manner*” [19].

These bureaucratic processes, including lengthy tendering procedures and complex administrative requirements, further complicate the already strained radiotherapy service in Gauteng province. Streamlining procurement practices and increasing transparency could help address these challenges and improve access to life-saving treatment in the region. While streamlining procurement practices is crucial, the impact of these bureaucratic challenges extends beyond acquisition to the operational realities within radiotherapy facilities [20,21]. The lack of timely maintenance and repairs, often a direct result of these inefficiencies, creates significant operational hurdles that further impede service delivery.

Moreover, beyond staffing and procurement challenges, healthcare professionals expressed significant concerns about operational issues stemming from poor communication, lack of teamwork, and collaboration within the radiation oncology multidisciplinary departments. One interviewee noted that there are siloed approaches within these departments, with insufficient coordination between key personnel like ROs, RTTs, and MPs, often leading to suboptimal treatment planning and delivery. This observation aligns with findings, highlighting the prevalence of communication breakdowns between these professional groups and their negative impact on patient care and echoing the concerns raised in a study by Selby et al. regarding the fragmentation of care in multidisciplinary oncology settings [5,22]. Another interviewee’s comment, “*We are constantly putting out fires because of communication breakdowns*”, highlights the urgency of addressing these issues. As Buchman et al. suggest, fostering a culture of interdisciplinary collaboration through regular multidisciplinary team meetings and targeted training programmes can significantly improve communication, teamwork, and, ultimately, patient outcomes in radiation oncology service delivery [23]. These challenges are not unique to Gauteng. Studies show that disparities in access to radiotherapy services persist, especially in low- and middle-income countries. Proposed solutions include task-shifting, which may improve radiotherapy access and reduce waiting times in the region.

This study contributes to the growing body of evidence [14,24,25,26] demonstrating the negative impact of limited radiotherapy capacity on cancer outcomes in LMICs. The long waiting times experienced by patients in Gauteng province due to insufficient facilities may lead to disease progression and reduced treatment efficacy, ultimately exacerbating existing health disparities. The potential consequences of limited radiotherapy access to patient outcomes are highlighted. Exploring alternative service delivery models, such as task-shifting, may offer potential strategies to improve radiotherapy access and reduce waiting times in Gauteng province. These strategies will include the framework that will be developed following this phase of the postgraduate research study aimed at improving access to radiotherapy and must consider the unique contextual factors, including economic and health system constraints, that shape radiotherapy service delivery in the region.

This study employed a convenience sampling method, which, while practical, introduces potential limitations regarding the generalizability of the findings. As detailed in the methodology section, delays in obtaining gatekeepers’ permission resulted in the exclusion of six private practices. This exclusion could potentially bias the sample, particularly if these practices share common characteristics that differ from the included practices. For example, if the excluded practices were predominantly high-volume centres, our findings might underestimate the actual utilisation of HEUs in the private sector. Additionally, one practice closed during the data collection period, further limiting the sample size and potentially affecting the representativeness of the private sector. Finally, one practice declined participation due to a busy schedule. While understandable, this non-response could introduce selection bias, as busier practices might have different resource allocation strategies or staffing patterns compared to less busy practices. These limitations should be considered when interpreting the study’s findings and generalising them to the broader radiotherapy landscape in Gauteng province. Future research with a larger, more representative sample is needed to confirm these findings and explore the nuances of HEU utilisation and workload across diverse practice settings.

## 5. Conclusions

In conclusion, addressing the challenges in Gauteng’s radiotherapy services, including bureaucratic inefficiencies, human resource shortages, and operational constraints, necessitates a comprehensive, multifaceted approach encompassing policy reforms, targeted investments, and robust public–private collaboration. This research underscores the urgent need for such interventions to improve access and equity in radiotherapy for cancer patients in Gauteng province.

To build upon these findings, the next phase of this research will focus on developing an actionable strategic framework. This framework will integrate quantitative data on time intervals, qualitative insights into patient experiences, and structural quality indicators to provide a practical roadmap for improving radiotherapy services. By synthesising results from all phases, this framework aims to guide decision-making and resource allocation, ensuring that interventions are evidence-based and patient-centred.

Ultimately, this research seeks to contribute to a more equitable and efficient cancer care system, not only in Gauteng but also in similar LMICs facing comparable challenges. By translating research into actionable strategies, we can strive to ensure that all patients have timely access to the life-saving benefits of radiotherapy, regardless of their socioeconomic status or geographic location.

## Figures and Tables

**Table 1 ijerph-22-00585-t001:** Data collection tool.

Data Collected	Variable
C1 Average number of patients per day (January–December 2023)	Patient/Day
C2 Average number of patients per month (January–December 2023)	Patient/Month
C3 Average number of patients awaiting radiotherapy per month (January–December 2023)	Radiotherapy
D1 Number of linear accelarators	Accelerators
D2 Number of CT simulators	Simulation
D3 Number of brachytherapy units	Brachytherapy units
D4 Number of MRIs	MRI
D5 Number of PET scan	PET Scan
E1 Number of personnel (RO)	RO
E2 Number of personnel (MP)	MP
E3 Number of personnel (RTT)	RTT
E4 Number of personnel (ON)	ON

Abbreviations: RO = radiation oncologist; MP = medical physicist; RTT = radiation therapist; ON = oncology nurse.

**Table 2 ijerph-22-00585-t002:** Descriptive analysis of public (N = 2) and private (N = 11) radiotherapy facilities.

Variables	Public Facilities					Private Facilities				
	Mean	Mode	SD ^1^	Min ^2^	Max ^3^	Mean	Mode	SD	Min	Max
Patient/Day	102.50	100 ^a^	3.54	100	105	31.18	12	12.86	12	52
Patient/Month	179.50	179 ^a^	0.71	179	180	45.91	15	31.56	15	125
Radiotherapy	115.00	56 ^a^	83.44	56	174	3.73	0	9.32	0	30
Accelerators	4.50	4 ^a^	0.71	4	5	1.27	1	0.47	1	2
Simulation	1.00	1	0.01	1	1	1.00	1	0.01	1	1
Brachytherapy units	1.00	1	0.01	1	1	0.27	0	0.47	0	1
MRI	0.50	0 ^a^	0.71	0	1	0.27	0	0.47	0	1
PET Scan	0.50	0 ^a^	0.71	0	1	0.09	0	0.30	0	1
RO	3.00	2 ^a^	1.44	2	4	4.18	3	2.18	1	9
MP	6.50	6 ^a^	0.71	6	7	1.45	1	0.52	1	2
RTT	22.00	21 ^a^	1.41	21	23	6.27	3 ^a^	2.61	3	11
ON	15.50	15 ^a^	0.71	15	16	0.18	0	0.41	0	1

^1^ SD—Standard deviation.; ^2^ Min—Minimum. ^3^ Max—Maximum. ^a^ Multiple modes exist. Abbreviations: RO = radiation oncologist; MP = medical physicist; RTT = radiation therapist; ON = oncology nurse.

**Table 3 ijerph-22-00585-t003:** High energy unit (linear accelerator) utilisation in public and private radiotherapy centres.

Centre ID	Centre Type (Public/Private)	Total Patients Treated/Year (Average)	Number of HEUs	HEU Utilisation Ratio (Patients/HEU) ^1^
1	Private	600	1	600
2	Private	540	2	270
3	Private	504	1	504
4	Private	744	2	372
5	Private	804	1	804
6	Private	276	1	276
7	Public	2160	4 (3) ^2^	540 (720)
8	Private	432	1	432
9	Private	180	1	180
10	Private	300	1	300
11	Private	180	1	180
12	Private	1500	2	750
13	Public	2148	5 (3) ^2^	430 (716)

^1^ Reference range (patients/HEU) 200–500; ^2^ number of HEUs in use.

**Table 4 ijerph-22-00585-t004:** Normalised linear accelerator availability in public and private sectors (linear accelerators per 1000 patients).

Sector	Number of Linear Accelerators	Patients Treated	Linear Accelerators per 1000 Patients
A ^1^	14	6060	2.31
B ^1^	9 (6) ^2^	4308	2.09 (1.39)

^1^ Sector A = private; B = public; ^2^ LINACs in use.

**Table 5 ijerph-22-00585-t005:** Workload ratio for public and private radiotherapy facilities.

	Centre Type (Public/Private)	Average Patients Treated/Year	Number of Radiation Oncologists	Number of Radiation Therapists	Number of Medical Physicists	Radiation Oncologist Workload (Patients/Year) ^1^	Radiation Therapist Workload (Patients/Year) ^2^	Medical Physicist Workload (Patients/Year) ^3^
1	Private	600	3	4	1	200	150	600
2	Private	540	3	7	2	180	77	270
3	Private	504	7	9	2	72	56	252
4	Private	744	5	11	2	149	68	372
5	Private	804	4	7	2	201	115	402
6	Private	276	4	5	1	69	55	276
7	Public	2160	4	21	7	540	103	309
8	Private	432	4	6	1	108	72	432
9	Private	180	1	3	1	180	60	180
10	Private	300	3	5	1	100	60	300
11	Private	180	3	3	1	60	60	180
12	Private	1500	9	9	2	167	167	750
13	Public	2148	2	23	6	1048	93	358

^1^ Recommended: 250–300. ^2^ Recommended: 100–150 ^3^ Recommended: 300–400.

## Data Availability

All data in this study are provided in the main manuscript.

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
