# Peer review of "Improving Access to Radiotherapy: Exploring Structural Quality Indicators for Radiotherapy in Gauteng Province, South Africa"

_ijerph, 2025, doi:10.3390/ijerph22040585_

Round 1
Reviewer 1 Report
Comments and Suggestions for Authors The paper explores the accessibility of radiotherapy resources in Gauteng Province, South Africa, using a mixed-methods approach. By combining quantitative data and qualitative interviews, it analyzes the disparities in radiotherapy resource allocation between public and private healthcare institutions. The study holds significant practical and policy implications, providing data support for improving the distribution and utilization of radiotherapy resources. However, there is still room for improvement in terms of research methodology, data analysis, and result interpretation. Add a discussion on sampling bias in the methodology section and consider how to mitigate the representativeness issues caused by convenience sampling. Optimize the structure of the results section to make data analysis more systematic and highlight the relationships between different variables. Enhance the integration of quantitative and qualitative data to improve the overall logical coherence and data support of the study. Include more specific policy recommendations in the discussion section, incorporating international experiences to provide more feasible improvement measures. Supplement the study with international research to enhance its global impact. Add some recent healthcare-related references
Comments on the Quality of English Language
The English could be improved to more clearly express the research.
Reviewer 2 Report
Comments and Suggestions for Authors
I read with interest this manuscript. The study idea is good and the methodology is well described. Moreover, the conclusions are in line with the study aim. I suggest to complete the manuscript with a brief section named "Future perspective" in that you could desscribe the future procedures and projects in order to improve the radiotherapy services in your Country.
Reviewer 3 Report
Comments and Suggestions for Authors
Minor grammatical and formatting edits throughout. Please see attached marked up manuscript.
The overall data interpretation and conclusions make sense but could be clarified with some extra explanation in some areas and also by normalizing the data sets to per patient or per X number of patients to make data easier to compare between the public and private sector groups.

Comments on the Quality of English Language
Minor suggestions throughout. Please see attached marked up copy for editorial suggestions.
Round 2
Reviewer 1 Report
Comments and Suggestions for Authors
The authors have already solved all of my problems.
Comments on the Quality of English Language
The English could be improved to more clearly express the research.